# Feeling safe or falling through the cracks— Patients' experiences of healthcare in cirrhosis illness: A qualitative study

Maria Hjorth [1,2,¤]*, Anncarin Svanberg[2‡], Daniel Sjöberg[1‡], Fredrik Rorsman[2‡], Elenor Kaminsky[3]

**1** Centre for Clinical Research in Dalarna, County of Dalarna, Falun, Sweden, **2** Department of Medical Sciences, Uppsala University, Uppsala, Sweden, **3** Department of Public Health and Caring Sciences, Uppsala University, Uppsala, Sweden

☙ These authors contributed equally to this work.
¤ Current address: Centre for Clinical Research, County of Dalarna, Falun, Sweden
‡ AS, DS and FR also contributed equally to this work.
* maria.hjorth@regiondalarna.se

**Data Availability Statement:** Data contain potentially identifying or sensitive patient information, sharing data may violate participant confidentiality in conflict to the Swedish Research

## Abstract

### Introduction

Patients with cirrhosis have a long-lasting relationship with medical personnel. Hierarchy in the healthcare contacts and feeling stigmatised may affect the patient's interactions with these care providers. Despite healthcare professionals' awareness of patients' increased self-care needs, patients report getting insufficient information and support. The patients' expectations and experiences of interacting with healthcare professionals in cirrhosis care is hence a research area that needs further investigation.

### Purpose

To capture patients' descriptions of healthcare experiences in relation to cirrhosis illness.

### Material and methods

Data comprise semi-structured interviews (N = 18) and open-ended questionnaire responses (N = 86) of patients with cirrhosis. Braun and Clarke's thematic analysis process was used, including both semantic and inductive elements. The study is reported following the COREQ guidelines.

### Findings

The analysis resulted in two themes: 1) Struggle to be in a dialogue and 2) Being helped or harmed. Six sub-themes were identified concerning aspects of experiences within each theme during the analysis. These sub-themes included: 'getting information', 'being involved', 'being perceived as a person', 'enduring care', 'feeling lost in the healthcare organisation', and 'not being taken care of'.

Ethical Review Authority approval. Data are available from Region Dalarna upon reasonable request (e-mail: forsknings. utlamnande@regiondalarna.se), provided that the data can be made available in accordance with applicable data protection and privacy regulations.

**Funding:** The first author, MH, was supported by Centre for Clinical Research, Falun– Uppsala University. The funders had no role in study design, data collection and analysis, decision to publish, or preparation of the manuscript.

**Competing interests:** The authors have declared that no competing interests exist.

## Conclusions

Patients with cirrhosis express concerns regarding where to turn in the continuum of cirrhosis care. They emphasise the importance of being involved in the dialogue with the healthcare professional, to be perceived as a person with a unique need to be informed. The healthcare organisation and continuity of care are either viewed as confusing or as helping to shape a safe and trustful contact, which was an important difference in feeling helped or harmed. Hence, patients wished for improved collaboration with healthcare professionals and to receive increased information about their disease. Person-centred communication in nurse-led clinics may increase patient satisfaction and prevent patients from falling through the cracks.

## Introduction

Cirrhosis is the end stage of long-standing chronic liver inflammation [1, 2]. The prevalence of cirrhosis is about 800 per 100,000 inhabitants [1], resulting in about one million deaths annually worldwide [2]. The occurrence of cirrhosis is deemed to increase due to harmful alcohol consumption and obesity [1]. In the early stages of the disease, symptoms are vague; moreover, cirrhosis is usually not diagnosed until the first episode of decompensation, i.e. the occurrence of ascites, hepatic encephalopathy and/or bleeding from gastroesophageal varices [3, 4]. A decompensation episode vastly implies an increased need for healthcare [5].

Patients with cirrhosis require a long-lasting relationship with healthcare professionals (HCPs). As for other chronic diseases, patients emphasise the importance of a trustful and continuous relation with HCPs [6, 7]. However, patients report receiving scarce information regarding disease and disease management from HCPs [6, 8–10]. Furthermore, the terminologies used by HCPs in their communication may hinder patients from understanding the information and making shared decisions. Regardless of the cirrhosis aetiology, but due to preconceptions that cirrhosis is solely caused by alcohol [5, 11, 12], patients experience stigmatisation, i.e. being ashamed, feeling guilty or feeling judged [5]. Consequently, patients may hesitate to seek healthcare, resulting in inadequate care and support [11]. Other patient-related factors, such as individual understanding and use of health information (health literacy) [13] and hepatic encephalopathy [14], may inhibit memory and learning abilities [13, 14].

In the treatment of cirrhosis, previously, patient care has mainly been physician-based [2], including general practitioners (GPs) and gastroenterologists or hepatologists at outpatient and inpatient care. Lately, however, involvement of registered nurses (RNs) has been suggested to improve the quality of care by providing patient information and self-care recommendations to prevent further cirrhosis complications [2, 15]. In the shift to involve nurses in cirrhosis care, it is important to understand the patients' experiences and opinions to identify gaps in the physician-based care, which nurses may complement. In our previous study regarding the lived experiences of patients with cirrhosis, participants spontaneously shared their experiences of physician-based healthcare [5].

The overall worldwide organisational goal, according to the World Health Organisation [16], is to provide high-quality care that is effective, efficient, accessible, acceptable, equitable and safe. From the patients' perspective, quality of care is influenced by their overall experiences of health, disease and healthcare services [17]. The HCP's ability to meet patients' expectations, needs and wishes thus also affects the patient's sense of quality. Therefore, healthcare

expertise, in contrast to patient's vulnerability from being sick, might cause a hierarchy between the two, which may have an impact on self-care [18]. In contrast, positive experiences from HCP meetings may improve patient adherence to treatment plans [7]. Patient-HCP communication, collaboration and management are thus central determinants of quality of care [19]. Consequently, quality of care is influenced by organisational and personal factors in both HCPs and patients. In order to improve clinical outcomes in Swedish healthcare, the patient's right to participate and collaborate has been clarified in the legislation [20]. Further, strategic national initiatives have been taken to implement person-centred care (PCC) [21].

Although the patients' perspective of care [17], including the conversation with HCPs [19], is an important determinant of quality of care [17, 19], studies regarding cirrhosis care are lacking. Previous reports disclose a need for improved support and information in cirrhosis care [8–10, 12]. Nonetheless, how cirrhosis patients experience collaboration and communication with HCPs as well as whether the care was individually adjusted according to patients' needs [21] remain unexplored. A complicating factor in cirrhosis is the looming sense of stigmatisation [5, 11], which implies that experiences from other patient populations [6, 7] cannot be directly transferred to the cirrhosis population. Accordingly, this study, which is part of a larger project on quality of care in patients with cirrhosis in an extended population [22], aims to capture patients' descriptions of healthcare experiences in relation to cirrhosis illness in a Swedish context.

## Materials and methods

### Study design

The study had an interpretative, descriptive design, with an inductive qualitative approach. The qualitative data encompassed 18 individual semi-structured interviews and 86 questionnaire responses [23, 24]. Data were analysed using Braun and Clarke's six step thematic analysis process, including both inductive and semantic elements [25].

### Sample and setting

Cirrhosis care in Sweden includes medical treatment at primary care, and outpatient and inpatient care units (Fig 1). Patients are usually treated within outpatient care; nonetheless, during disease decompensation, inpatient and, sometimes, intensive care is required. Some patients are referred to highly specialised centres for a liver transplantation evaluation. Occasionally, cirrhosis comorbidities are managed in primary care or by other medical specialists.

Patients diagnosed with cirrhosis were recruited via medical records for individual interviews [5], and among patients that participated in a nurse-led intervention [22]. Informants were selected by purposive maximum variation sampling for interviews [5]. Eighteen patients at two hepatology outpatient clinics in mid-Sweden (one university and one rural hospital) spontaneously described their healthcare experiences in the continuum of cirrhosis care (Fig 1). One hundred sixty-eight patients were enrolled and underwent stratified randomisation to the nurse-led intervention study [22] at six hospitals (four university and two rural hospitals) in mid- and south-Sweden. Eighty-six questionnaires, where at least one of two open-ended questions was answered, were included. The remaining 82 questionnaires, where some of the responses were blank (n = 62) or the responses did not correspond to the study's aim (n = 20), were excluded. All identifiable patient information was exchanged with study codes.

Severity of cirrhosis was classified with the Child-Pugh score [26]. Child-Pugh score A implies mild cirrhosis symptoms, whereas Child-Pugh score B or C indicates disease progression with aggravated symptoms. Transition from Child-Pugh A to B/C delineates increased risk of liver-related mortality. Hepatic encephalopathy, a neuropsychiatric disorder, may occur

| Where | Primary care | ⟷ | Hospital - outpatient care | ⟷ | Hospital - inpatient care | ⟷ | Highly specialised care |
|---|---|---|---|---|---|---|---|
| **What** | Liver disease diagnosis | | Liver cirrhosis diagnosis | | Acute liver cirrhosis decompensation | | Liver transplant assessment |
| | Treatment of comorbidity | | Liver biopsy | | Emergency room | | Liver transplant surgery |
| | Preventive care | | Secondary prevention | | X-ray examination | | |
| | Home care | | Surveillance programme | | Intensive care | | |
| | | | Laparocentesis | | Intubation and/or inotropic drugs | | |
| | | | Gastroscopy | | Liver transplant investigation | | |
| | | | Liver transplant investigation | | | | |
| **Who** | General practitioner | | Internal medicine physician, gastroenterologist or hepatologist | | Internal medicine physician, gastroenterologist or hepatologist | | Hepatologist |
| | Telephone-line to registered nurse | | Telephone-line to registered nurse | | Emergency care personnel | | Transplant surgeon |
| | Addictiive behavioural therapist | | Dietitian | | Registered nurse | | Transplant coordination nurse |
| | | | Endoscopist | | Endoscopist | | Dietitian |
| | | | | | Dietitian | | Physiotherapist |
| | | | | | Physiotherapist | | Anesthesiologist |
| | | | | | Radiologist | | Radiologist |
| | | | | | Anesthesiologist | | |
| | | | | | Infection specialist | | |

**Fig 1. The continuum of cirrhosis care in Sweden.**

in discrete or non-clinical grade to overt symptoms [27]. All patients in this study underwent neuropsychological tests, the portosystemic encephalopathy (PSE) paper-and-pencil battery of tests and/or continuous reaction time (CRT) [28, 29], before enrolment. Patients with overt hepatic encephalopathy were excluded, whereas patients with discrete or non-clinical symptoms were accepted. No informants had been invited to the nurse-led intervention study before the interview. However, after the interviews, six of the interviewed patients later also participated in the nurse-led intervention and replied to the open-ended questions in the questionnaire. Patients were included after receiving I) the information letter, II) oral information via telephone and III) providing written informed consent. Details regarding inclusion criteria for interviews and questionnaires are described in detail elsewhere [5, 22]. Informants' characteristics are presented in Table 1.

## Data collection

Data comprised 18 interview transcripts and answers to 86 open-ended questions. The 18 interviews were conducted during 2016–2017 at their local hospital. The interviewer did not ask about healthcare experiences. Instead, the informants spontaneously reported their healthcare experiences at the same time as they talked about their experiences of day-to-day life in relation to cirrhosis illness. For these individuals, patient care was part of life and was described in broad variations. The semi-structured interviews were conducted face-to-face, as described elsewhere [5]. The electronic questionnaire 'Quality of care from the patients' perspective' [23, 24] was answered at the local outpatient clinic, without the presence of healthcare staff or relatives, upon enrolment in a nurse-led intervention study during 2016–2020 [22]. The two open-ended questions, answered by 86 participants, refer to experiences from the hepatology outpatient care as follows: 'I was particularly pleased with this' and 'suggestions for improvements'. There were no prior contacts between the informants and the first author who

**Table 1. Characteristics of informants from interviews and questionnaires.**

|  | Classification | Informants | |
|---|---|---|---|
|  |  | Interviews (N = 18) | Questionnaires (N = 86) |
| Gender | Men | 9 | 52 |
|  | Women | 9 | 34 |
| Age | 18–39 | 2 | 1 |
|  | 40–64 | 10 | 45 |
|  | 65–85 | 6 | 40 |
| Marital status | Single | 3 | 35 |
|  | Cohabiting | 15 | 51 |
| Level of education | None | 1 | 0 |
|  | Elementary school | 3 | 18 |
|  | Upper secondary school | 9 | 40 |
|  | University | 5 | 28 |
| Employment | Student/working | 7 | 22 |
|  | Sick leave | 2 | 10 |
|  | Retired | 5 | 40 |
|  | Disability pension | 2 | 4 |
|  | Unemployed | 0 | 5 |
|  | Other | 2 | 5 |
| Child-Pugh score | A | 12 | 54 |
|  | B | 4 | 26 |
|  | C | 2 | 6 |
| Aetiology of liver disease | Alcohol | 3 | 44 |
|  | Hepatitis B/C | 2 | 4 |
|  | Primary Biliary Cholangitis | 1 | 0 |
|  | Primary Sclerosing Cholangitis | 3 | 0 |
|  | Autoimmune hepatitis | 2 | 5 |
|  | Non-alcoholic steatohepatitis | 5 | 15 |
|  | Cryptogenic | 1 | 12 |
|  | ≥ 2 aetiologies | 1 | 6 |
| Time since cirrhosis diagnosis | <1 year | 2 | 69 |
|  | 1–5 years | 8 | 17 |
|  | >5 years | 8 | 0 |
| Hepatic encephalopathy* | None | 11 | 56 |
|  | Discrete/non-clinical | 7 | 27 |

*missing data n = 3

conducted the interviews, nor the RNs that recruited participants for the nurse-led self-care programme.

## Data analysis

The rich interview descriptions and the data from the 98 participants covering the current study's aim were considered appropriate for a certain level of interpretation in the analysis. To report the core of participants' experiences, a thematic analysis, including both inductive and semantic approaches, was performed in six systematic steps according to Braun and Clarke [25]. First, the authors familiarised themselves with the data by reading interview transcripts

repeatedly, while making notes. Second, significant text features were assigned a code to describe the data patterns generously. Third, codes were organised into tentative themes, and questionnaire data were analysed according to steps 1 to 3. Fourth, themes were evaluated iteratively against codes and text transcriptions, being visualised on a thematic map. Fifth, the themes were interpreted, defined and labelled. Finally, the results were reported based on the important messages, exemplified by quotes. The semantic approach enabled objective systematisation with interpretation of themes regarding informants' healthcare experiences in relation to cirrhosis illness. The initial analysis was performed by the first author (MH) in close collaboration with the last author (EK), who has extensive experience in qualitative analysis. The analysis was continuously discussed with all authors to reach a consensus.

The Consolidated criteria for Reporting Qualitative Research guidelines (COREQ) checklist provided guidance in reporting the study [30]. Rigour was ensured through the strategies of trustworthiness, including confirmability, credibility, dependability and transferability [31]. Conformability implies objective analysis, which was facilitated through three research group members with, and two without, experience from hepatology (inpatient and outpatient care). Credibility concerns the applicability of the analysis, based on the aim and interpretation of the data. The semantic approach for the thematic analysis aided the close descriptions of the informants' experiences. Using a thematic map and informants' quotations to illustrate the findings further increased interpretation and transparency from the empirical raw data. For dependability, the authors strived for a transparent and logical description of the research process. To facilitate transferability, extensive variables relating to the informants' characteristics were disclosed (Table 1). Moreover, the authors sought to describe the study's settings and context, as well as the data collection in detail, in the method section. Following a qualitative approach, our purpose was not to quantify participants' responses. NVivo software [NVivo qualitative data analysis software; QSR International Pty Ltd. Version 20, 2021] was used to sort the data.

## Theoretical framework

A useful framework for how to practise person-centred care is Santana's conceptual framework, including the three domains of structure, process and outcome [32]. This is used for a discussion of the study's results, and hence further explained here. *Structure* includes the organisational system facilitating person-centred care. The philosophical grounds are patients' collaboration and rights, which imply that care shall be delivered with respect and in agreement. Furthermore, organisational supporting structures for e-health and measurements of person-centred care performance should be included. The healthcare environment should also be welcoming to patients, i.e. having a convenient design. *Process* describes the patient-HCP interaction, i.e. the HCP's role in achieving person-centred care in four aspects: cultivating communication, respectful and passionate care, engaging patients in their care, and integration of care. The four aspects include the HCP's skills in being able to grasp and share information from the patient's perspective and needs. Moreover, HCPs should practise an empathetic approach with respect to psychological and cultural needs, and viewing the patient as an expert. The collaboration comprises shared decision-making regarding goals, self-care and care-plans. Furthermore, coordination of care, communication and referrals in-between HCPs as well as continuity of care are emphasised. *Outcome* refers to the availability of, and timely access to, care and patient-reported outcomes, such as health-related quality of life and patients' experience of healthcare.

## Ethical considerations

The study followed the principles of the Helsinki Declaration [33]. The Swedish Ethics review authority approved the study (2016/146). Confidentiality was ensured by assigning a code to

each participant. Participation was voluntary and could be interrupted whenever the participants wished, without giving a reason.

## Findings

Research data comprised 18 interviews and 86 questionnaire responses. This contributed to comprehensive data regarding patients' experiences of up to ten years of cirrhosis illness. Fifty-six patients (n = 7 interviewed; n = 49 surveyed) had experience of decompensation, i.e. ascites, hepatic encephalopathy or oesophageal varices. Men were in the majority (58%), and alcohol-related cirrhosis was the predominant diagnosis (45%). Age varied from 24 to 83 (mean 62). The majority (63%) presented mild cirrhosis symptoms, consistent with Child-Pugh score A, at the time of data collection (Table 1).

### Themes

The analysis resulted in two themes, describing patients' healthcare experiences about their cirrhosis illness: 'Struggle to be in a dialogue' and 'being helped or harmed'. Six sub-themes established a pattern of different aspects of the experiences within each theme (Fig 2). The findings are described under theme headings in the text below, with sub-themes written in italics, illustrated by quotes.

**Struggle to be in a dialogue.** Although some patients expressed satisfaction with the information received, several had difficulties in *getting information* according to their expectations during outpatient care visits. The physicians' engagement, competence and interest in cirrhosis affected what type of information the patients received.

> *Another doctor at the Gastrointestinal department. . . he never took me in; he just said it's good, it's good. But [my new doctor]. . . took care of me. . . said there is something wrong. . . I was so tired*

(Informant 2, cirrhosis diagnosis 1–2 years)

In addition, the way the physician provided information affected how it was understood. Receiving answers to one's questions was reassuring, whereas having questions unanswered was perceived as being ignored by the HCP. Written information about cirrhosis illness was desired but rare. Ambiguous information conveyed in conversations or written communication, such as letters or one's medical records, made patients draw their own conclusions or feel

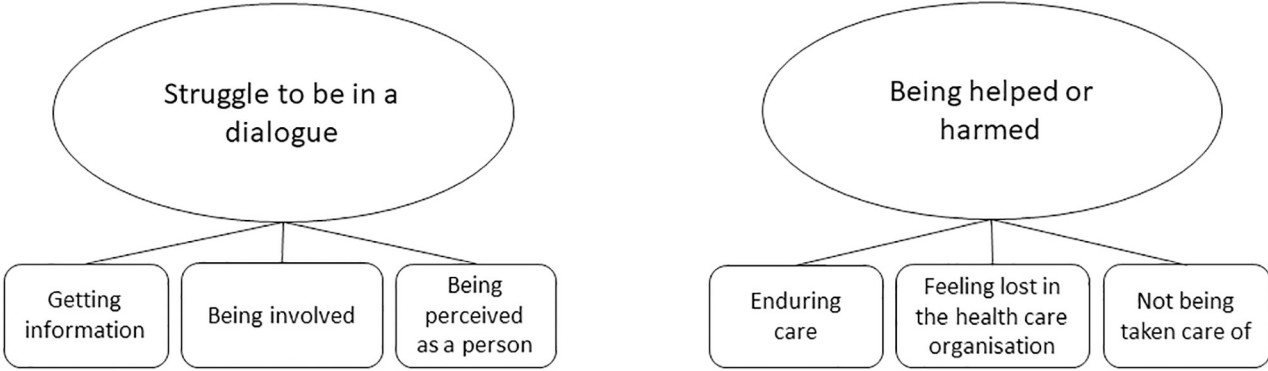

**Fig 2. Study themes and sub-themes regarding patients' healthcare experiences regarding their cirrhosis illness.**

anxious. Another obstacle for receiving adequate information was the time limit for the physician's appointment. Consequently, patients lacked knowledge about cirrhosis disease and prognosis, examinations, treatment and self-care. To compensate for one's dissatisfaction, some patients asked to change to another physician. Other patients suggested increased appointment times to receive expanded information.

> *You want more answers. . . When they did an ultrasound of the liver, so,. . . "[it] was the same finding as before, no change". . . maybe, I want a different answer. . . that they should develop it a little more.*

(Informant 5, cirrhosis diagnosis 2–4 years)

In the outpatient care, patients both desired and actively strived to *be involved* in their care, which was reflected in a large quantity of the study's data, including reflections on one's own attributes and their interaction with HCPs. Some patients could become involved by communicating and describing their concerns, by being brave enough to ask important questions and standing up for oneself. This was exemplified by taking command in the HCP meeting:

> *The doctor is sitting there to help me and not vice versa. . . it's my body, and I can ask what I want about it. It's me that needs to have the answer. If I don't understand, I have to ask again.*

(Informant 12, cirrhosis diagnosis 5–10 years)

These patients were empowered to set requirements for their care. For example, they pronounced their own free will in making choices regarding, e.g. how to undergo examinations; whether to stop drinking alcohol; to decide which physician they wanted; and whether to have contact with HCPs at all. In contrast, the patient's personal skills and attributes could sometimes make involvement difficult, e.g. being forgetful, not daring to be honest, not wanting to nag, or being uncertain about how to ask questions.

> *It's hard to be open. . . to be honest. . . you have to lie and it's awful. I want. . . to be able to sit and talk to my doctor and say. . . "I buy a pack of beer. . . sometimes". . . but I don't want to say that. "No, damn it, I don't drink" I say. . . although I may have been drinking. . . I. . . want. . . to be able to be honest. . . without getting reprisals for it, you are almost punished. . . to have a beer every now and then. . . I can't mention THAT.*

(Informant 18, cirrhosis diagnosis > 10 years ago)

When the HCP was encouraging, it was easier to ask questions. The patients were unaware of the requirement of patient consent regarding whether they wanted to undergo a liver transplantation. Therefore, some patients found the question unnecessary, as this was the only option for survival.

In the HCP interaction, patients felt involved when given the opportunity to follow one's laboratory tests—to see an improvement in the tests was a source of joy. Being able to influence the frequency of visits to one's physician made patients feel secure, i.e. having regular annual outpatient appointments, but with the possibility of having an earlier appointment within short notice when needed. Their sense of security was decreased when the HCP did not take the patient's wishes into account. Agreements between the patient and physician were important for patients to establish trust and be satisfied with their physician's competence in decisions regarding their treatment. Hence, when an agreement was not followed, the patient felt

disappointed, e.g. neglected feedback on laboratory tests in-between appointments. On some occasions, patients' desires for their care were not fully met. This was exemplified as having doubts about the GP's recommendations; patients had frequently asked the GPs to consult with the responsible hepatologist, which was a request that was accepted reluctantly by GPs. Some patients had not been invited to participate in the patient-physician dialogue. Occasionally, the dialogues were experienced as standardised, with limited time and lack of interest in the questions:

> *I could. . . talk. . . discuss a little more. . . if I got the chance but. . . I. . . am given an assembly line. . . They want a standard unit. . . The doctors don't know how to handle a patient who is interested in the way I am. . . I [have] requested. . . and received appointments a couple of times; of course, he talks to me but I. . . ask questions that he might not be so comfortable with.*

(Informant 1, cirrhosis diagnosis > 10 years ago)

In contacts with the outpatient care, patients expected to *be perceived as a person*. The HCP relation created a sense of safety via a professional and personal HCP approach, with dialogue based on a positive spirit and empathy. Furthermore, confirmation by the physician, through respect and attention, and having their own decisions and actions acknowledged, were important factors for feeling respected as a person with unique needs. However, in the outpatient care, patients found it difficult to receive care covering the overall perspective of their health. Instead, some patients described fragmented care when consulting physicians, as they lacked a holistic view:

> *They are so heavily specialised and. . . it's good in a way, but then, they can. . . nothing but THAT. Then, they know nothing about anything else.*

(Informant 9, cirrhosis diagnosis > 10 years ago)

When the patient's privacy was overlooked in the communication, they felt misunderstood, insulted, or as if they were being accused or reprimanded by the HCPs. Some described feelings as though they were falling through the cracks when communication went right over their heads, when they were not listened to, or their symptoms were ignored.

> *When I was in the hospital. . . then they. . . came in and wore the worst protective clothing. Then I asked why. . . it was because I had. . . hepatitis. Yes, but I said I have an autoimmune disease, there is nothing that is contagious, but. . . they did not know. . . Then you really felt infected with the plague, but you know that you are not.*

(Informant 5, cirrhosis diagnosis 2–4 years)

**Being helped or harmed.** The cirrhosis illness forced patients to *endure care*, which affected them both physically and psychologically. Receiving care was thus not only about getting help, but it was also a threat to their own well-being. For some patients, being in the hospital setting evoked a feeling of wanting to escape. Patients regularly had examinations such as liver biopsy, gastroscopy or magnetic resonance imaging. These investigations caused discomfort to the patients, both during and after the procedures, i.e. claustrophobia, pain or loss of integrity. In anticipation of a scheduled examination or blood test, patients started worrying about the procedure and results. Moreover, the hospital environment itself was perceived as

unpleasant due to, for example, poor standard at the premises or ventilation. In addition, healthcare visits reminded patients of the sometimes silent liver disease:

*It's like a big insecurity that's there all the time. . .like it's gnawing there. . .in the subconscious. . .You can. . .keep it at bay most of the time. . .Those moments. . .when you come into contact with healthcare, that's when you're really reminded of it. Otherwise, you're pretty good at pushing it away.*

(Informant 15, cirrhosis diagnosis 5–10 years)

Patients had a *feeling of being lost in the healthcare organisation* and therefore wished for continuity in care, e.g. meeting the same physician at every appointment, and help in coordinating their healthcare contacts. It was tiring being sick and simultaneously having to navigate in the healthcare organisation, e.g. making calls to order prescriptions, or booking or cancelling appointments.

*In one month. . . there are 6–7 different doctor visits, and you need to have contact with the pharmacy. . . you should call. . . pick up the medicine. . . "but you have no prescription left for" aha well, then I will call the doctor. . . and order the medicine . . . Keep track of it. . . it's like. . . a damn chase all the time, I think. . . You [should] go to a doctor [-visit]. . . and then no, damn. . . have. . . damn pain. . . swollen and awful, so you have to call. . . and cancel your appointment, otherwise you have to pay the cost. . . So, it's damn hard to be sick.*

(Informant 18, cirrhosis diagnosis > 10 years ago)

When patients contacted outpatient care, they expected to receive specific health care related to cirrhosis. Sometimes, they had other health-related problems or serious symptoms, which resulted in them often being re-directed to other HCPs, such as GPs or the emergency ward. This made patients feel confused and insecure, since they wanted to discuss their illness with a hepatology specialist. Patients felt relieved when physicians took the initiative to discuss their case with colleagues when necessary:

*Then. . . he. . . said himself that. . . "I will talk to [a colleague]. . . Then we will see what we come up with". . . Then. . . it becomes. . . easier for me, instead of having to sort of go on in different places.*

(Informant 4, cirrhosis diagnosis 5–10 years)

The outpatient care was perceived as fragmented since patients themselves had to request extra appointments with the physician or other HCPs, such as RNs or social workers. These resources were not provided in a structured manner; rather, the physician, wanting to be accommodating, agreed to the patient's wish.

Although the majority of patients reported having positive experiences with physicians in outpatient care, in the continuum of care, there were situations where patients felt *not being taken care of*. To avoid feelings of being neglected, factors such as being taken seriously and receiving care on time were deemed important. Some patients described that it literally took years until they received an appointment for further investigation, when they were first identified as having high liver values following a blood test. After developing cirrhosis, they realised that this delay had resulted in inaccurate care, which caused disappointment. Due to the GP's limited knowledge, the perspectives of the cirrhosis illness were sometimes disregarded, or the GP suggested inappropriate treatment.

*I asked [my GP], "can I drink alcohol?". . . I read online that it was not good to combine [with the medicine]. . . he said. . . "it doesn't matter. . . most people who take this. . . drink alcohol, it's not a problem". [I] said "my liver values are a bit . . . high., "Most people have high", he said . . . So I was very blessed by this doctor . . . that I did not have to change my lifestyle in any way . . . In hindsight, I think that was wrong.*

(Informant 1, cirrhosis diagnosis > 10 years ago)

In the outpatient care, the long waiting times to receive one's examination results and to make an appointment with a physician made patients anxious. Despite a scheduled contact with their physician, some visits had not been conducted accordingly. When investigations involved both outpatient and highly specialised care, patients sometimes experienced long periods without any contact, which was mentally distressing:

*It takes a long time. . . Sometimes, it's mentally difficult. . . It. . . has, then taken so many years. Not much has happened.*

(Informant 6, cirrhosis diagnosis 1–2 years)

The patients' disappointment in being treated incorrectly and concerns during waiting times made them wish for improved support.

## Discussion

This study aimed to capture descriptions of patients' healthcare experiences when having cirrhosis illness, which resulted in two main themes: 'Struggle to be in a dialogue' and 'being helped or harmed'. Generally, patients trusted and felt safe with the care received by HCPs for cirrhosis. A striking finding was patients' great need for collaboration and thorough dialogue with HCPs, which requires that patients are involved and listened to, in order to become an equal partner in their care, and feel understood. Furthermore, the patient's personal attributes and confidence were important factors for daring to engage in the care relationship. According to the Swedish Patient Act [20], patients have the right to be involved in their care. However, the findings disclose communication barriers, which inhibit patient involvement. These barriers were not due to the lack of HCPs' medical knowledge, but because of the fact that the communication itself was inadequate. As previously reported, the patients experienced stigmatisation in their day-to-day life [5], which could also influence their relationship with the HCP [11]. To our knowledge, an imbalanced patient-HCP relation has not previously been described among the cirrhosis population, but in other patient populations with chronic illnesses [18]. Hence, in line with Avallin et al. [19], HCPs need to especially facilitate patient collaboration by adapting communication skills, which constitutes the hallmark of being taken care of.

The patients' limited cirrhosis knowledge caused them to feel worried and insecure. Congruent with Low et al. [8], the patients in the present study perceived the information provided by HCPs as being too medicalised. Information thus seemed to be given without regard to the patient's ability to understand [13]. Therefore, to compensate for patients' low level of knowledge on cirrhosis [8–10, 12], simply conveying information may not be sufficient. An incomprehensible language complicates the management of one's cirrhosis illness [8, 12] and has a negative impact on the psychological well-being [8]. Consistent with previous reports on cirrhosis populations in the United Kingdom [8, 34], the United States [35] and Denmark [12], our patients stated that short and infrequent visits make it difficult to learn about the disease. Altogether, in line with Avallin et al. [19], our findings emphasise that the vocabulary used by

HCPs, their ability to capture the patient's wishes and how the information was provided, created a sense of security. The findings highlight that patients' desire to receive care support was individual and influenced by previous experiences, self-efficacy and knowledge. Hence, in contrast to Valery et al. [9], the need for HCP support did not only follow disease progression or the burden of symptoms. In line with Waibel et al. [7], our data support that patients need HCP continuity and information about where to turn to in-between appointments to avoid the feeling of falling through the cracks. HCPs need to develop strategies to communicate the unpredictable trajectory of cirrhosis in a timely manner [8], training in how to increase patient involvement in cirrhosis care [32], and also to adapt the conversation based on the person's understanding [13]. As one step towards improving communication, we propose that HCPs need to ask patients about their expectations, their previous understanding of the situation and supplement verbal information with written information.

The patient experience of being lost in the healthcare system is a previously undocumented finding in the continuum of cirrhosis care. The fragmented care and long waiting times resulted in patients feeling disappointed, concerned and as if they had been forgotten. Hence, patients with cirrhosis have similar experiences to those with other chronic diseases, e.g. chronic heart failure [6]. Moreover, the continuum of cirrhosis care was confusing for patients, as they were unsure of where to go when feeling worse. Overall, this expands our previous report on the unpredictable and unsafe situation for patients with cirrhosis [5]. Therefore, we suggest that patients should have stable and continuous contact with HCPs specialising in cirrhosis. Furthermore, patients need to receive thorough guidance on 'who, how and when' to contact when cirrhosis care is necessary. Consequently, HCPs need to be perceptive of the patient's unique situation and needs in a person-centred care fashion [36]; therefore, the present findings will be further discussed using Santana's theoretical framework of person-centred care [32].

The *structure* domain of Santana's framework was reflected in patients' descriptions of the hospital premises and the environment, which were unpleasant and difficult to navigate. For patients, reading their electronic medical records was confusing and sometimes frightening. Healthcare inconveniences, which were endured and narrated by patients in the present study, may, according to Sinclair et al. [37], be prevented and alleviated by a compassionate HCP. Concerning the *process* domain, most patients confirmed having a compassionate and trustful HCP relation. However, undoubtedly, improvements are required in the areas of providing information to the patients and replying patients regarding examinations or laboratory. In line with person-centred frameworks, our patients emphasised the importance of being listened to and being involved in decisions regarding their care. Hence, HCPs should strive for patient participation and provide support based on the individual needs [17, 19, 32]. Our finding of receiving insufficient patient information has also been reported in cancer care [38] and chronic heart failure care [6], and thus seems to be a general issue in chronic disease care. Regarding the third domain, *outcome*, our patients, in general, expressed satisfaction with outpatient care for cirrhosis, nevertheless, they gave examples of areas for improvement, e.g. waiting times to get test results and individualised information. In the findings, there are several contradictions, e.g. regarding the satisfaction with perceived information or the ability to be involved. The differences further justify the application of person-centred care because the patient's expectations and resources may differ. One example of a clinical implementation of person-centred care proceeds from the patient's narrative and the partnership, which involves carefully listening to frame the patient's unique situation [21]. Furthermore, joint decisions form the basis for the co-creation of the care plan, which is continuously updated and documented. Patient education provided by HCPs trained in person-centred communication [21, 32] may help to improve cirrhosis care [2, 15]. A person-centred and multi-professional care

may optimise patient satisfaction and self-care performance, which is currently implemented and evaluated by nurse-led clinics within cirrhosis outpatient care [22].

## Strengths and limitations

This is the first study emphasising the need for patient involvement in cirrhosis care. Due to the lack of knowledge in this area, we found an interpretive, descriptive, inductive qualitative design appropriate to answer the study's aim without any pre-constructed theories [25]. The purposive sampling may involve sampling bias, as patients feeling stigmatised or having negative healthcare experiences might have declined participation. Patients both desired and actively strived to be involved in their care. Although the interview questions did not intend to explore patients' healthcare experiences, data included rich experiences to answer the current study's aim. Moreover, during the interviews, patients addressed their experiences of cirrhosis care repeatedly. The fact that informants were not prompted to answer specific questions is a weakness that may prevent all perspectives, such as stigmatisation, of the patients' care experience from being captured. Hence, in future studies, it is important to ask specific questions regarding healthcare experiences. Given this limitation, a discussion regarding data saturation was not applicable. Although only 86 of the 168 questionnaire responses contributed with answers to the present study's aim, the responding patients represented all six hospitals that were involved. Several participants had previous experience with a higher Child Pugh class, i.e. B or C. The result thus reflects experiences of different stages of cirrhosis. The two datasets together reflect a representative population of cirrhosis, i.e. predominantly male (n = 61) and alcohol-related cirrhosis (n = 47) (Table 1) [3, 4]. Since no quantification was made in this qualitative study, the six informants who contributed to interviews and questionnaires were not considered to influence the findings negatively. The authors strived for a detailed description of empirical claims regarding trustworthiness, i.e. confirmability, credibility, dependability, and transferability [31]. To avoid the influence of pre-understanding in the data analysis, the first author, with long experience in hepatology, worked closely with the last author, who lacked this experience. The concepts of objectivity and confirmability were thus met, in addition to theoretical, methodological, and analytical considerations. Throughout the analysis process, codes, theme descriptions, and their coherence with raw data content were discussed in several internal research meetings and external review seminars. The iterative work during the analysis confirmed the findings in the transcribed text, which encouraged credibility. Further, the choice of a semantic approach in the thematic analysis, with certain interpretations of the patients' experiences and use of quotations, aimed at increasing objectivity for confirmability and also credibility, through a clear link to the raw data. The researchers have aimed to provide clear descriptions at each step of the analysis process to demonstrate dependability. The context of the study may limit the transferability of our findings to cirrhosis healthcare in Sweden. Nonetheless, in line with previous research [6, 8–10, 12], our findings also emphasised patients' great need to receive information and need for care coordination. The extensive sample (Table 1), together with the continuum of Swedish cirrhosis care, as described (Fig 1), may increase the possibility of transferability to other national or international contexts with similar healthcare organisations.

## Conclusions

Patients' healthcare experiences in relation to their cirrhosis illness were summarised in two themes: 'Struggle to be in a dialogue' and 'Being helped or harmed'. For patients with cirrhosis, healthcare sometimes felt uncertain, imperfect, and difficult to navigate. Patients highlighted the importance of being involved in the HCP dialogue, to be perceived as a person with a

unique need for information. The healthcare organisation and continuity of care could either be experienced as confusing or it created a safe and trustful HCP contact, which was an important difference in being helped or harmed. The shared experiences imply that caring was more than just serving patients with care. Henceforth, HCPs trained in person-centred communication may provide individualised information to increase patients' involvement in decisions, encourage self-care management, and satisfaction with cirrhosis care. In the continuum of cirrhosis care, RNs may provide person-centred information and well-coordinated care in order to prevent patients from feeling as though they are falling through the cracks. Future cirrhosis nurse-led clinics should be designed to meet patients' demands of holistic and structured outpatient healthcare.

## Acknowledgments

The authors acknowledge all informants, who so enthusiastically shared their experiences.

## Related manuscripts

Hjorth M, Sjoberg D, Svanberg A, Kaminsky E, Langenskiold S, Rorsman F. Nurse-led clinic for patients with liver cirrhosis-effects on health-related quality of life: study protocol of a pragmatic multicentre randomised controlled trial. BMJ Open. 2018;8(10):e023064.

Hjorth M, Svanberg A, Sjöberg D, Rorsman F, Kaminsky E. Liver cirrhosis turns life into an unpredictable roller coaster: A qualitative interview study. Journal of Clinical Nursing. 2020;29(23–24):4532–43.

## Author Contributions

**Conceptualization:** Maria Hjorth, Anncarin Svanberg, Daniel Sjöberg, Fredrik Rorsman, Elenor Kaminsky.

**Data curation:** Maria Hjorth.

**Formal analysis:** Maria Hjorth, Anncarin Svanberg, Daniel Sjöberg, Fredrik Rorsman, Elenor Kaminsky.

**Funding acquisition:** Maria Hjorth.

**Investigation:** Maria Hjorth.

**Methodology:** Maria Hjorth, Elenor Kaminsky.

**Project administration:** Maria Hjorth.

**Resources:** Maria Hjorth, Daniel Sjöberg, Fredrik Rorsman.

**Software:** Maria Hjorth.

**Supervision:** Anncarin Svanberg, Daniel Sjöberg, Fredrik Rorsman, Elenor Kaminsky.

**Validation:** Maria Hjorth, Elenor Kaminsky.

**Visualization:** Maria Hjorth.

**Writing – original draft:** Maria Hjorth.

**Writing – review & editing:** Maria Hjorth, Anncarin Svanberg, Daniel Sjöberg, Fredrik Rorsman, Elenor Kaminsky.

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
