## [Decision Letter · Decision Letter 0]

11 Sep 2022

PONE-D-22-11597Well received or falling through the cracks – patients’ health care experiences with liver cirrhosis illness: A qualitative studyPLOS ONE

Dear Dr. Hjorth,

Thank you for submitting your manuscript to PLOS ONE. After careful consideration, we feel that it has merit but does not fully meet PLOS ONE’s publication criteria as it currently stands. Therefore, we invite you to submit a revised version of the manuscript that addresses the points raised during the review process.

Please note that we have only been able to secure a single reviewer to assess your manuscript. We are issuing a decision on your manuscript at this point to prevent further delays in the evaluation of your manuscript. Please be aware that the editor who handles your revised manuscript might find it necessary to invite additional reviewers to assess this work once the revised manuscript is submitted. However, we will aim to proceed on the basis of this single review if possible.  The reviewer has concerns that the discussion is repeating the results without presenting new information, that the section applying person-centred care would benefit from a re-worked analysis, and has concerns about whether the data presents a large enough variation in patients’ health care experiences in relation to LC. Please address these, and all, concerns.

We look forward to receiving your revised manuscript.

Kind regards,

Alice Coles-Aldridge

Editorial Office

PLOS ONE

Journal Requirements:

2. We noted in your submission details that a portion of your manuscript may have been presented or published elsewhere. Please clarify whether this  publication was peer-reviewed and formally published. If this work was previously peer-reviewed and published, in the cover letter please provide the reason that this work does not constitute dual publication and should be included in the current manuscript.

Reviewers' comments:

Reviewer's Responses to Questions

**Comments to the Author**

1. Is the manuscript technically sound, and do the data support the conclusions?

Reviewer #1: Yes

2. Has the statistical analysis been performed appropriately and rigorously? 

Reviewer #1: N/A

3. Have the authors made all data underlying the findings in their manuscript fully available?

Reviewer #1: Yes

4. Is the manuscript presented in an intelligible fashion and written in standard English?

Reviewer #1: Yes

5. Review Comments to the Author

Reviewer #1: PONE-D-22-11597 Well received or falling through the cracks – patients’ health care experiences with liver cirrhosis illness: A qualitative study

1. The study presents the results of original research.

Yes, this is original research.

2. Results reported have not been published elsewhere.

Yes

3. Experiments, statistics, and other analyses are performed to a high technical standard and are described in sufficient detail.

High technical standard and in sufficient detail. Perhaps clarify how many interviews and questionnaire in one sentence at the data collection and place the sentence on raw 151 as the first sentence at data analysis—In total….

Please clarify the 6 steps in the analysis to show that there was TA analysis, as it is presented now it is very similar to content analysis. For example, initial codes are produced from data in step 2 and data is coded into so many patterns as possible. Etc…

The aim was to describe patients’ health care experiences in relation to liver cirrhosis illness.

The title is; Well received or falling through the cracks – patients’ health care experiences with liver cirrhosis illness: A qualitative study

Data collection, more information is supplied in reference 3.- so this is a secondary analysis of that interview data? In reference 3 data was analysed with content analysis- was there enough data to make a secondary analysis, with another perspective and with some interpretation attached? Please, clarify. The interview questions in reference 3 have focus on life situation, how could data respond to health care experiences?

Trustworthiness is presented, according to reference 30.

The result is two themes, Being engaged, collaborating and feeling well-received and Feeling nervous, lost and receiving insufficient information. The result presentation is clearer if the sub-themes are left out- presenting only two themes are more story telling according to TA (thematic analysis). One reflection here, these two themes seem to be two sides of the same coin and they are just piled sub-themes labels- they are not presenting the meaning or the content of the theme (according to TA)

I think that there is a need to re-work the analysis, the information in data seem to be about a struggle to be in dialogue (getting information, being involved, and perceived as a person) and there is uncertainty about the health care system (ignorance about the disease, no continuity and not being taken care of). Re-working the analysis should strengthen the result.

This study is focusing on an important and interesting area of research, so if the analysis were analysed according to TA there would be interesting information.

The discussion is repeating the result and confirmed by references used, nothing new presented, but this could be due to the “directed analysis”. The section applying person-centred care is interesting and would benefit of a re-worked analysis.

Limitations are well presented but I am not sure that data presents a large variation of patients’ health care experiences in relation to LC. This section could be shortened- it is repeating some information and some information should be placed at data collection section etc.

4. Conclusions are presented in an appropriate fashion and are supported by the data.

Conclusions are presented in appropriate fashion, but it is a repeated result in short-short version. Even so, there are some highlights about uncertainty and so on, but this is not mentioned in the result….

5. The article is presented in an intelligible fashion and is written in standard English.

Yes, the article is presented in an intelligible fashion, and it is written in standard English. The introduction is well-written and structured, leading me as a reader forward to the aim.

6. The research meets all applicable standards for the ethics of experimentation and research integrity.

Yes, this study meets all the applicable standards for research integrity.

7. The article adheres to appropriate reporting guidelines and community standards for data availability.

Yes, the article is following the reporting guidelines.

This could be an interesting paper, presenting important knowledge. This paper needs to be re-analysed then it would be suitable for publication.

Out of 37 references 10 are 10 years old or more, there are 23 references in the introduction and out of these 5 are 10 years old or more (mostly instruments references).

6. PLOS authors have the option to publish the peer review history of their article (what does this mean?). If published, this will include your full peer review and any attached files.

Reviewer #1: No

---

## [Author Response · Author response to Decision Letter 0]

25 Oct 2022

Dear Editor and Reviewer,

Thank you for the comments in order to improve our manuscript 'Well received or falling through the cracks - patients health care experiences with liver cirrhosis illness: A qualitative study'.

According to your suggestion we have reanalysed our data. Further we have strived to improve our manuscript according to each of the reviewer comments. We hope our manuscript now meet the criteria for publication in PLOSONE. In the file 'Response to Reviwers' we have answered to each of the aspects the reviewer found important, we hope you find our reponses and improvements sufficiently described.

Best regards,

The research group

---

## [Decision Letter · Decision Letter 1]

12 Dec 2022

PONE-D-22-11597R1Feeling safe or falling through the cracks – patients’ health care experiences with liver cirrhosis illness: A qualitative studyPLOS ONE

Dear Dr. Hjorth,

Thank you for submitting your manuscript to PLOS ONE. After careful consideration, we feel that it has merit but does not fully meet PLOS ONE’s publication criteria as it currently stands. Therefore, we invite you to submit a revised version of the manuscript that addresses the points raised during the review process.

The manuscript has been evaluated by three reviewers, and their comments are available below.

The reviewers have raised a number of concerns that need attention. They request additional information on methodological aspects of the study (such as how the participants were selected and specifics of the theoretical approach taken with respect to the qualitative analyses), clarification in the results section (including which data are from which set of participants), and request revisions to the introduction and discussion sections.

Could you please revise the manuscript to carefully address the concerns raised?

We look forward to receiving your revised manuscript.

Kind regards,

Steve Zimmerman, PhD

Associate Editor, PLOS ONE

Reviewers' comments:

Reviewer's Responses to Questions

**Comments to the Author**

1. If the authors have adequately addressed your comments raised in a previous round of review and you feel that this manuscript is now acceptable for publication, you may indicate that here to bypass the “Comments to the Author” section, enter your conflict of interest statement in the “Confidential to Editor” section, and submit your "Accept" recommendation.

Reviewer #1: All comments have been addressed

Reviewer #2: (No Response)

Reviewer #3: (No Response)

2. Is the manuscript technically sound, and do the data support the conclusions?

Reviewer #1: (No Response)

Reviewer #2: Yes

Reviewer #3: No

3. Has the statistical analysis been performed appropriately and rigorously? 

Reviewer #1: (No Response)

Reviewer #2: N/A

Reviewer #3: N/A

4. Have the authors made all data underlying the findings in their manuscript fully available?

Reviewer #1: (No Response)

Reviewer #2: No

Reviewer #3: No

5. Is the manuscript presented in an intelligible fashion and written in standard English?

Reviewer #1: (No Response)

Reviewer #2: Yes

Reviewer #3: No

6. Review Comments to the Author

Reviewer #1: Thank you for your efforts in amending and clarifying the manuscript. All comments have been taken into consideration.

I am looking forward to see it published

Reviewer #2: PONE-D-22-11597R1

The authors tackle an important area of study in cirrhosis care. In order for the healthcare system to provide more patient-centered care, studies such as this one are needed to focus our care improvement problem. While I was not a primary reviewer, I do see that the authors have significantly edited the paper. Despite this, I would recommend further revisions based on my comments below.

General comments:

1. “Being helped or harmed” – not sure this theme name carries a meaning that is obvious; perhaps something about Benefits and Burden of care since the authors include subthemes of burden of care.

2. This is my personal bias, but the term liver cirrhosis is redundant and just cirrhosis should be used.

3. The term “caregiver” is often used in the literature to refer to a family member who provides supplementary care. Healthcare provider or healthcare professional would be a more proper terms for this paper.

Methods:

1. How were the 20 patients selected for individual interviews?

Results:

1. Informant CPT score or class should be included (in addition to the number of years with cirrhosis)

Discussion:

1. The authors on a few occasions talk about stigma and reluctance to seeking care in cirrhosis – is this a finding of their study? If not and is more a reference to prior literature, I would advise the authors minimize this as their own findings show that patient did not feel stigmatized or show reluctance to seek care (likely due to sample bias in this case).

2. Discussion really needs to be more succinct and focus on what a clinician should take a way to change practice while emphasizing the 2 themes they found. Additionally, the focus should be on authors think healthcare providers need to accomplish change in practice – education, training, e-tools, handouts. As an example, line 353-354: “However, the findings disclose communication barriers that inhibit patients’ right to be involved in the conversation [23].” Is this a finding of their analysis or reference 23. Right now, the first 2 paragraphs of the discussion spend a lot of time reviewing the literature. It is hard to tell what is new from this study and what has already been shown. As such, the two paragraphs should be combined as both emphasize the need for “person-centered” communication and care and more streamlined. Lines 370-378 represent a more succinct and focused writing style where I walked away with a clear understanding of the results in the context of prior literature as well as what I should do differently.

3. Similarly, in the next paragraph - lines 362-367, what do the authors suggest as a change – longer visits, more educational material between visits, etc.

4. The paragraph before limitations/strengths is also long, somewhat redundant and again ends with patient-centered care. But how does one achieve this? A review of literature to answer this question would be much more relevant to the discussion. If there isn’t much literature in this area, then what do that authors recommend given their interview with patients as aims for future studies?

5. Strength and limitations section does not list any real limitations. What biases were introduced based on how patients were enrolled and selected for the detailed intereview? Since patients were not specifically asked about cirrhosis related healthcare experiences, did this introduce any bias – missed themes that patients would have discussed if prompted in a more structured fashion? How about response bias – of 168 patients enrolled, only 86 responses were used in the analysis? How about the fact this is just one center and one country - does it apply to other countries with other healthcare systems.

Reviewer #3: First of all, I would like to congratulate you on the work you have done. I believe that learning about the experience of living with chronicity is a great objective and that the current literature lacks this type of study, which is so necessary in order to give patients a voice and thus increase the quality of their care. However, I believe that there are certain issues that need to be improved in order for this study to be published.

Summary

In general it reflects the work done, however I think that if they included what type of caregivers they are referring to from the beginning it would better situate the reader. Also, the location of where the study was conducted should be included in the material and methods. Finally, the wording of your conclusions does not really correspond to the results of your study, so I would recommend that you rewrite this section taking into account your own data.

Introduction

In general, the whole introduction is very disorganised, jumping from one topic to another, without any thread of argument or order. I consider all the information they provide to be very relevant, as it situates the reader in the state of the question, but this disorder makes it difficult to read. As in the summary, I would recommend that they talk about professional carers. They also talk about some of the stigma attached to liver cirrhosis, but there is no depth and I think this is an important topic to delve into. As for including Santana's conceptual framework in the introduction, I think it is totally incorrect. This framework should be explained in the methodology section and the introduction should be used to correctly situate the reader as to what is known and what remains to be known. In fact, I believe that this is something that is missing in your introduction and is a clear justification of the need to know the experience of those affected from their own perspective. Furthermore, in their introduction they talk about the experience of chronicity in general, while later they focus their study on the patient-health professional relationship. Are there any studies that show that this relationship is not adequate? Why is there a need to listen to those affected themselves? I suggest first of all to put the ideas in this section in order and then to justify in a more concrete way the need for this study.

Material and Methods

This section raises many doubts in my mind. They talk about an interpretative, descriptive and inductive design, but these are characteristics that are already presupposed in a qualitative research, therefore, I do not understand the need to expose it. On the other hand, there is no mention at any point of qualitative theoretical foundations. Are we dealing with phenomenology, symbolic interactionism or do they intend to carry out a grounded theory study? I think that framing your study within one of the qualitative theoretical foundations would allow for a more robust design.

In terms of design it is confusing. First they do 18 semi-structured interviews and then they ask two open questions to 86 patients... What does this add? I think that with the 18 interviews there would be enough information to put together good results. So, this design is not correct and this makes the results inadequate. On the other hand, they talk about semi-structured interviews and in their script they do not include questions about health professionals? Or at least that is what they state in the data collection section.

Data analysis

They do not reflect the work done, nor the training of the people who carried out the analysis. Moreover, in my opinion, it is not necessary to explain Braun and Clarke's method so extensively and you should explain better how you adapted this method to your research.

Results

It is impossible for me to know which of your results correspond to N=18 or N=86? Assuming that they are from the semi-structured interviews, I think that these results are not well presented, as certain statements are made that do not follow from the data, and therefore this should be in the discussion section. Perhaps it is for this reason that the results should be reduced, as they should be unified and not repeated throughout the whole section. Furthermore, there are contradictions in her own results and this is not described later in the discussion.

Discussion: There is an important part of the discussion that is not clear from the results. For example, they talk about stigmatisation and this has not appeared in their results. They should rewrite it and emphasise the important issues. It would also be advisable to work more on the lack of training of health professionals in communication skills and to justify this lack of training with other studies that could confirm the results they have obtained.

In general, I consider that the objective is appropriate and the topic very interesting, but there are methodological errors that need to be improved to make the study more robust.

7. PLOS authors have the option to publish the peer review history of their article (what does this mean?). If published, this will include your full peer review and any attached files.

Reviewer #1: No

Reviewer #2: No

Reviewer #3: No

---

## [Author Response · Author response to Decision Letter 1]

3 Feb 2023

Dear Reviewers,

We are greatful for the time and efforts you have made to help us improve our manuscript "Feeling safe or falling through the cracks – patients’ health care experiences with liver cirrhosis illness: A qualitative study". In the response letter we have throughly responded to each of your suggestion, with references to the page and line of our improvements. We hope you consider the changes intelligable and satisfactory.

Best regards

the Authors

---

## [Decision Letter · Decision Letter 2]

13 Mar 2023

Feeling safe or falling through the cracks – patients’ experiences of healthcare in cirrhosis illness: A qualitative study

PONE-D-22-11597R2

Dear Dr. Hjorth,

We’re pleased to inform you that your manuscript has been judged scientifically suitable for publication and will be formally accepted for publication once it meets all outstanding technical requirements.

Kind regards,

Jason Scott

Academic Editor

PLOS ONE

Additional Editor Comments (optional):

Reviewers' comments:

Reviewer's Responses to Questions

**Comments to the Author**

1. If the authors have adequately addressed your comments raised in a previous round of review and you feel that this manuscript is now acceptable for publication, you may indicate that here to bypass the “Comments to the Author” section, enter your conflict of interest statement in the “Confidential to Editor” section, and submit your "Accept" recommendation.

Reviewer #1: All comments have been addressed

Reviewer #2: All comments have been addressed

2. Is the manuscript technically sound, and do the data support the conclusions?

Reviewer #1: Yes

Reviewer #2: Yes

3. Has the statistical analysis been performed appropriately and rigorously? 

Reviewer #1: N/A

Reviewer #2: Yes

4. Have the authors made all data underlying the findings in their manuscript fully available?

Reviewer #1: Yes

Reviewer #2: Yes

5. Is the manuscript presented in an intelligible fashion and written in standard English?

Reviewer #1: Yes

Reviewer #2: Yes

6. Review Comments to the Author

Reviewer #1: Thank you for your efforts in once again amending and clarifying the manuscript. All comments

have been taken into consideration.

I am looking forward to see it published

Reviewer #2: The authors have addressed my major concerns with significant changes throughout the manuscript to improve flow and highlight the results of the study.

7. PLOS authors have the option to publish the peer review history of their article (what does this mean?). If published, this will include your full peer review and any attached files.

Reviewer #1: No

Reviewer #2: No

---

## [Editor Report · Acceptance letter]

23 Mar 2023

PONE-D-22-11597R2 

Feeling safe or falling through the cracks – patients’ experiences of healthcare in cirrhosis illness: A qualitative study 

Dear Dr. Hjorth:

I'm pleased to inform you that your manuscript has been deemed suitable for publication in PLOS ONE. Congratulations! Your manuscript is now with our production department. 

Kind regards, 

on behalf of

Dr. Jason Scott 

Academic Editor

PLOS ONE